# Blurring Structure and Learning to Optimize and Adapt Receptive Fields

## Abstract

The visual world is vast and varied, but its variations divide into structured and unstructured factors. We compose free-form filters and structured Gaussian filters, optimized end-to-end, to factorize deep representations and learn both local features and their degree of locality. In effect this optimizes over receptive field size and shape, tuning locality to the data and task. Our semi-structured composition is strictly more expressive than free-form filtering, and changes in its structured parameters would require changes in architecture for standard networks. Dynamic inference, in which the Gaussian structure varies with the input, adapts receptive field size to compensate for local scale variation. Optimizing receptive field size improves semantic segmentation accuracy on Cityscapes by 1-2 points for strong dilated and skip architectures and by up to 10 points for suboptimal designs. Adapting receptive fields by dynamic Gaussian structure further improves results, equaling the accuracy of free-form deformation while improving efficiency.

## 1 Introduction

Although the visual world is varied, it nevertheless has ubiquitous structure. Structured factors, such as scale, admit clear theories and efficient representation design. Unstructured factors, such as what makes a cat look like a cat, are too complicated to model analytically, requiring free-form representation learning. How can recognition harness structure without restraining the representation?

Free-form representations are structure-agnostic, making them general, but not exploiting structure is computationally and statistically inefficient. Structured representations like steerable filtering (Freeman & Adelson, 1991; Simoncelli & Freeman, 1995; Jacobsen et al., 2016), scattering (Bruna & Mallat, 2013; Sifre & Mallat, 2013), and steerable networks (Cohen & Welling, 2017) are efficient but constrained to the chosen structures. We propose a new, semi-structured compositional filtering approach to blur the line between free-form and structured representations and learn both. Doing so learns local features and the degree of locality.

Free-form filters, directly defined by the parameters, are general and able to cope with unknown variations, but are parameter inefficient. Structured factors, such as scale and orientation, are enumerated like any other variation, and require duplicated learning across different layers and channels. Nonetheless, end-to-end learning of free-form parameters is commonly the most accurate approach to complex visual recognition tasks when there is sufficient data.

Structured filters, indirectly defined as a function of the parameters, are theoretically clear and parameter efficient, but constrained. Their effectiveness hinges on whether or not they encompass the true structure of the data. If not, the representation is limiting, and subject to error. At least, this is a danger when *substituting* structure to replace learning.

We *compose* free-form and structured filters, as shown in Figure 1, and learn both end-to-end. Free-form filters are not constrained by our composition. This makes our approach more expressive, not less, while still able to efficiently learn the chosen structured factors. In this way our semi-structured networks can reduce to existing networks as a special case. At the same time, our composition can learn different receptive fields that cannot be realized in the standard parameterization of free-form filters. Adding more free-form parameters or dilating cannot learn the same family of filters. Figure 2 offers one example of the impracticality of architectural alternatives.

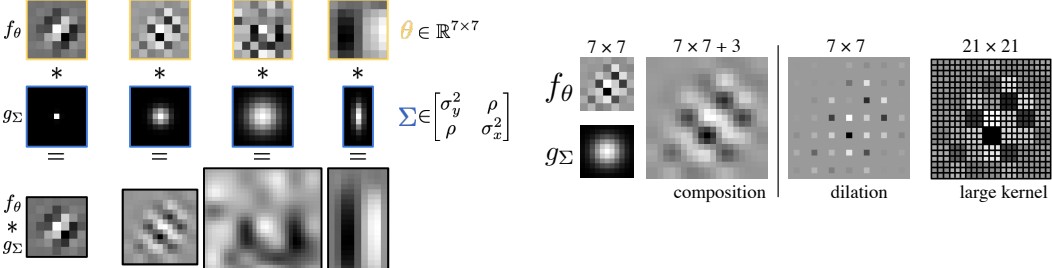

Figure 1: We compose free-form filters $f_\theta$ and structured Gaussian filters $g_\Sigma$ by convolution $*$ to define a more general family of filters than either alone. Our composition makes filter size differentiable for end-to-end learning.

Figure 2: Our composition does not reduce to dilation or more free-form parameters. Dilation is sparse, causing artifacts, and cannot be learned. More free-form parameters require more data to learn and the maximum size is still bounded.

Our contributions include: (1) composing Gaussian filtering and free-form filtering to bridge classic ideas for scale-space representation design and current practices for representation learning, (2) exploring a variety of receptive fields that our approach can learn, and (3) adapting receptive fields during inference with accurate and efficient dynamic Gaussian structure.

## 2 RELATED WORK

Composing structured Gaussian filters with free-form learned filters draws on structured filter design and representation learning. Our work is inspired by the transformation invariance of scale-space (Lindeberg, 1994), the parsimony of steerable filtering (Freeman & Adelson, 1991; Perona, 1995; Bruna & Mallat, 2013; Cohen & Welling, 2017), and the adaptivity of dynamic inference (Olshausen et al., 1993; Jaderberg et al., 2015; De Brabandere et al., 2016; Dai et al., 2017). Analysis that the effective receptive field size of deep networks is limited (Luo et al., 2016), and is only a fraction of the theoretical size, motivates our goal of learning and adapting unbounded receptive field sizes and varied receptive field shapes.

**Transformation Invariance** Gaussian scale-space and its affine extension connect covariance to spatial structure for transformation invariance (Lindeberg, 1994). We jointly learn structured transformations via Gaussian covariance and features via free-form filtering. Enumerative methods cover a set of transformations, rather than learning to select transformations: image pyramids (Burt & Adelson, 1983) and feature pyramids (Kanazawa et al., 2014; Shelhamer et al., 2017; Lin et al., 2017) cover scale, scattering (Bruna & Mallat, 2013) covers scales and rotations, and steerable networks (Cohen & Welling, 2017) cover discrete groups. Our learning and inferring covariance relates to scale selection (Lindeberg, 1998), as exemplified by the scale invariant feature transform (Lowe, 2004). Scale-adaptive convolution (Zhang et al., 2017) likewise selects scales, but without our Gaussian structure and smoothness.

**Steering** Steering indexes a continuous family of filters by linearly weighting a structured basis, such as Gaussian derivatives. Steerable filters (Freeman & Adelson, 1991) index orientation and deformable kernels (Perona, 1995) index orientation and scale. Such filters can be stacked into a deep, structured network (Jacobsen et al., 2016). These methods have elegant structure, but are constrained to it. We make use of Gaussian structure, but keep generality by composing with free-form filters.

**Dynamic Inference** Dynamic inference adapts the model to each input. Dynamic routing (Olshausen et al., 1993), spatial transformers (Jaderberg et al., 2015), dynamic filter nets (De Brabandere et al., 2016), and deformable convolution (Dai et al., 2017) are all dynamic, but lack local structure. We incorporate Gaussian structure to improve efficiency while preserving accuracy.

Proper signal processing, by blurring when downsampling, improves the shift-equivariance of learned filtering (Zhang, 2019). We reinforce these results with our experiments on blurred dilation, to complement their focus on blurred stride. While we likewise blur, and confirm the need for smoothing to prevent aliasing, our focus is on how to jointly learn and compose structured and free-form filters.

## 3 A CLEAR REVIEW OF BLURRING

We introduce our chosen structured filters first, and then compose them with free-form filters in the next section. While the Gaussian and scale-space ideas here are classic, our end-to-end optimized composition and its use for receptive field learning are novel.

### 3.1 GAUSSIAN STRUCTURE

The choice of Gaussian structure determines the filter characteristics that can be represented and learned. For learning, it is differentiable, low-dimensional for parameter efficiency, and expressive enough for different sizes and shapes. For signal processing, it is smooth and computationally efficient. In particular, the Gaussian has these attractive properties for our purposes:

- shift-invariance for convolutional filtering,

- normalization to preserve input and gradient norms for stable optimization,

- separability to reduce computation by replacing a 2D filter with two 1D filters,

- and cascade smoothing from semi-group structure to decompose filtering into smaller, cumulative steps.

In fact, the Gaussian is the unique filter satisfying these and further scale-space axioms (Koenderink, 1984; Babaud et al., 1986; Lindeberg, 1994).

The Gaussian kernel in 2D is $G(\mathbf{x}; \mathbf{\Sigma}) = \frac{1}{2\pi\sqrt{\det \mathbf{\Sigma}}} e^{-\mathbf{x}^T \mathbf{\Sigma}^{-1} \mathbf{x}/2}$ for input coordinates $x$ and covariance $\Sigma \in \mathbb{R}^{2 \times 2}$, a symmetric positive-definite matrix.

The structure of the Gaussian is controlled by its covariance $\Sigma$. Note that we are concerned with the spatial covariance, where the coordinates are considered as random variables, and not the covariance of the feature dimensions. Therefore the elements of the covariance matrix are $\sigma_y^2$, $\sigma_x^2$ for the y, x coordinates and $\rho$ for their correlation. The standard, isotropic Gaussian has identity covariance $\begin{bmatrix} 1 & 0 \\ 0 & 1 \end{bmatrix}$. There is progressively richer structure in spherical $\begin{bmatrix} \sigma^2 & 0 \\ 0 & \sigma^2 \end{bmatrix}$, diagonal $\begin{bmatrix} \sigma_y^2 & 0 \\ 0 & \sigma_x^2 \end{bmatrix}$, and full $\begin{bmatrix} \sigma_y^2 & \rho \\ \rho & \sigma_x^2 \end{bmatrix}$ covariances which can represent scale, aspect, and orientation in one, two, or three parameters.

Selecting the right covariance yields invariance to a given spatial transformation (Lindeberg, 1994). We leverage this transformation property of Gaussians to learn receptive fields in Section 4.1 and dynamically adapt them for locally invariant filtering in Section 4.2.

From the Gaussian kernel $G(x, \Sigma)$ we instantiate a Gaussian filter $g_\Sigma(\cdot)$ in the standard way: (1) evaluate the kernel at the coordinates of the filter coefficients and (2) renormalize by the sum to correct for this discretization. We decide the filter size according to the covariance by setting the half size $= \lceil 2\sigma \rceil$ in each dimension. This covers $\pm 2\sigma$ to include $95\%$ of the true density no matter the covariance. (We found that higher coverage did not improve our results.) Our filters are always odd-sized to keep coordinates centered.

### 3.2 COVARIANCE PARAMETERIZATION & OPTIMIZATION

The covariance $\Sigma$ is symmetric positive definite, requiring proper parameterization for unconstrained optimization. We choose the log-Cholesky parameterization (Pinheiro & Bates, 1996) for iterative optimization because it is simple and quick to compute: $\Sigma = U'U$ for upper-triangular $U$ with positive diagonal. We keep the diagonal positive by storing its log, hence *log*-Cholesky, and exponentiating when forming $\Sigma$. (See Pinheiro & Bates (1996) for a primer on covariance parameterization.)

Here is an example for full covariance $\Sigma$ with $\sigma_y^2$, $\sigma_x^2$ for the y, x coordinates and $\rho$ for their correlation: $\begin{bmatrix} \sigma_y^2 & \rho \\ \rho & \sigma_x^2 \end{bmatrix} \leftarrow \begin{bmatrix} +1 & -2 \\ -2 & +8 \end{bmatrix} = \begin{bmatrix} +1 & +0 \\ -2 & +2 \end{bmatrix} \begin{bmatrix} +1 & -2 \\ +0 & +2 \end{bmatrix} = (\log(1), -2, \log(2))$. Spherical and diagonal covariance are parameterized by fixing $\rho = 0$ and tying/untying $\sigma_y, \sigma_x$.

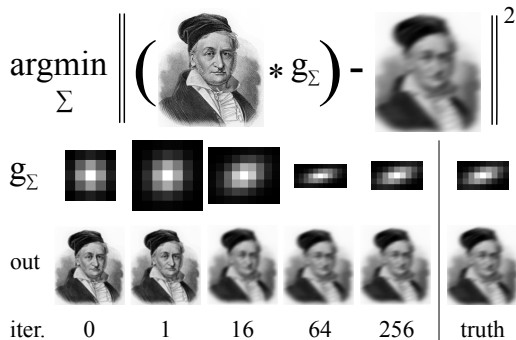

Figure 3: Recovering an unknown blur by optimizing covariance. Gradient optimization of the structured parameters $\Sigma$ quickly converges to the true Gaussian. Although this is a simple example, it shows how the Gaussian can represent scale, aspect, and orientation with variable filter size.

### 3.3 LEARNING TO BLUR

As a pedagogical example, consider the problem of optimizing covariance to reproduce an unknown blur. That is, given a reference image and a blurred version of it, which Gaussian filter causes this blur? Figure 3 shows such an optimization: from an identity-like initialization the covariance parameters quickly converge to the true Gaussian.

Given the full covariance parameterization, optimization controls scale, aspect, and orientation. Each degree of freedom can be seen across the iterates of this example. Had the true blur been simpler, for instance spherical, it could still be swiftly recovered in the full parameterization.

Notice how the size and shape of the filter vary over the course of optimization: this is only possible through structure. For a Gaussian filter, its covariance is the intrinsic structure, and its coefficients follow from it. The filter size and shape change while the dimension of the covariance itself is constant. Lacking structure, free-form parameterization couples the number of parameters and filter size, and so cannot search over size and shape in this fashion.

## 4 COMPOSING FREE-FORM & GAUSSIAN FILTERING

Deep visual representations are made by composing convolutions to learn rich features and *receptive fields*, which characterize the spatial extent of the features. Although each filter might be small, and relatively simple, their composition can represent and learn large, complex receptive fields. For instance, a stack of two $3 \times 3$ filters is effectively $5 \times 5$ but with fewer degrees of freedom ($2 \cdot 3^2$ vs. $5^2$). Composition therefore induces factorization of the representation, and this factorization determines the generality and efficiency of the representation.

Our semi-structured composition factorizes the representation into spatial Gaussian receptive fields and free-form features. This composition is a novel approach to making receptive fields differentiable, low-dimensional, and decoupled from the number of parameters. Our approach jointly learns the structured and free-form parameters while guaranteeing proper sampling for smooth signal processing. Purely free-form filters cannot learn size and shape in this way: size is bounded by the number of parameters and shape depends on all of the parameters. Purely structured filters, restricted to Gaussians and their derivatives for instance, lack the generality of free-form filters. Our factorization into structured and free-form filters is efficient for the representation, optimization, and inference of receptive fields without sacrificing the generality of features.

Receptive field size is a key design choice in the architecture of fully convolutional networks for local prediction tasks (Shelhamer et al., 2017). The problem of receptive field design is encountered with each new architecture, dataset, or task. Trying candidate receptive fields by enumeration is costly, in effort and computation, whether by manual or automated model search (Zoph & Le, 2017; Kandasamy et al., 2018; Liu et al., 2019). By making this choice differentiable, we show that learning can adjust to changes in the architecture and data in Section 5.2. Optimizing our semi-structured filters in effect searchs over receptive fields. Our composition helps relieve the burden of architecture design by relaxing the receptive field from a discrete decision into a continuous optimization.

## 4.1 Convolutional Composition of Static Gaussians

Our composition $f_\theta \circ g_\Sigma$ combines a free-form $f_\theta$ with a structured Gaussian $g_\Sigma$. This combination reduces to convolution, and so it inherits the efficiency of aggressively tuned convolution routines. Since convolution is associative, filtering of an input $I$ decomposes into two steps of convolution by $I * (g_\Sigma * f_\theta) = I * g_\Sigma * f_\theta$.

This decomposition has computational advantages. The Gaussian step can be done by specialized filtering that harnesses separability, cascade smoothing, and other structure. Memory can be spared by only keeping the covariance parameters and instantiating the filters as needed. Each compositional filter can always be explicitly formed by $g_\Sigma * f_\theta$ for visualization (see Figure 1) or other analysis.

Both $\theta$ and $\Sigma$ are differentiable for end-to-end learning.

There are useful special cases of the Gaussian when considering covariance optimization as differentiable model search. Figure 4 illustrates covariances for the identity, initialization, and pooling. Because the identity is included in the limit, our models can recover standard networks that lack our composition. By initializing near the identity, we are able to augment pre-trained networks without interference, and let learning decide whether or not to make use of structure.

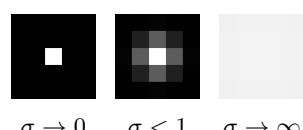

$\sigma \to 0 \quad \sigma < 1 \quad \sigma \to \infty$

Figure 4: The identity is recovered by a delta as variance goes to zero. Small variance gives a good initialization near the identity. Average pooling is approximated as variance goes to infinity.

**Blurring for Smooth Signal Processing** Even without learning the covariance, blurring can improve dilation by avoiding aliasing. Figure 5 shows the effect of blurring dilation. Smoothing when subsampling is a fundamental technique in signal processing (Oppenheim & Schafer, 2009), so we merely note this fix as a simple alternative to the careful re-engineering of dilated architectures (Yu et al., 2017; Wang et al., 2018).

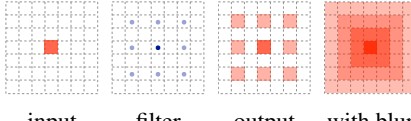

input    filter    output    with blur

Figure 5: Blurring prevents aliasing by smoothing dilated filters to respect the sampling theorem.

**Compound Gaussian Structure** Gaussian filters have a special compositional structure: cascade smoothing. Composing a Gaussian $g_\Sigma$ with a Gaussian $g_{\Sigma'}$ is still Gaussian with covariance $\Sigma + \Sigma'$. This lets us efficiently assemble *compound* receptive fields made of multiple Gaussians. Center-surround (Kuffler, 1953) receptive fields, which boost contrast, can be realized by such a combination as Difference-of-Gaussian (Rodieck & Stone, 1965) (DoG) filters, which subtract a larger Gaussian from a smaller Gaussian. Our joint learning of their covariances tunes the contrastive context of the receptive field, extending (Ding et al., 2018) which learns contrastive filters with fixed receptive field sizes.

**Design Choices** We cover the design choices involved in the application of our composition. As a convolutional composition, it can augment any convolution layer in the architecture. We focus on including our composition in late, deep layers to show the effect without much further processing. We add compositional filtering to the output and decoder layers of fully convolutional networks because the local tasks they address rely on the choice of receptive fields.

Having decided where to compose, we must decide how much structure to include. We explore minimal structure, where one Gaussian is shared across the free-form filters of a layer, and spatially dynamic structure, where the receptive field at each location varies with the input.

## 4.2 Deformable Composition of Dynamic Gaussians

*Dynamic* inference replaces static, global parameters with local parameters, inferred from the input, to adapt to these variations. There are two routes to dynamic Gaussian structure: local filtering and deformable sampling. Local filtering has a different filter kernel for each position, as done by dynamic filter networks (De Brabandere et al., 2016). This ensures exact filtering for dynamic Gaussians, but is too computationally demanding for large-scale recognition networks. Deformable sampling adjusts the position of filter taps by dynamic offsets, as done by deformable convolution (Dai et al., 2017). We exploit deformable sampling for the sparse approximation of dynamic Gaussians.

We constrain deformable sampling to Gaussian structure by setting the sampling points according to covariance. Figure 6 illustrates Gaussian deformations. Our default deformation approximates the standard Gaussian. We consider the same progression of spherical, diagonal, and full covariance for dynamic structure. This low-dimensional structure differs from the high degrees of freedom in a dynamic filter network, which sets free-form filter parameters, and deformable convolution, which sets free-form offsets. Our Gaussian deformation requires only a small, constant number of covariance parameters independent of the sampling resolution and the kernel size $k$, while deformable convolution has constant resolution and requires $2k^2$ offset parameters for a $k \times k$ filter.

To infer the local covariances we learn a convolutional regressor, which is simply a convolutional filter. The inferred covariances then determine the receptive fields of the the following free-form filters. The low-dimensional structure of our dynamic parameters makes this regressor more efficient than the regressor for free-form deformation, as it only has three outputs for each full covariance, or even just one for each spherical covariance. Since the covariance is differentiable, the regression is learned end-to-end from the task loss without further supervision.

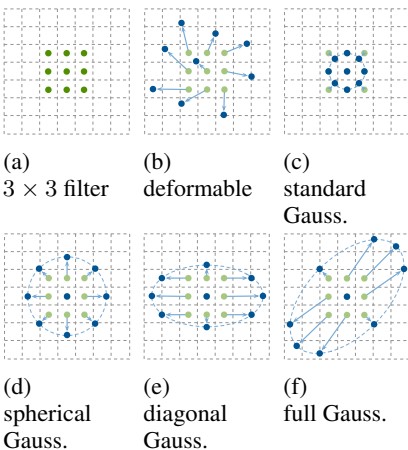

(a) $3 \times 3$ filter    (b) deformable    (c) standard Gauss.

(d) spherical Gauss.    (e) diagonal Gauss.    (f) full Gauss.

Figure 6: Gaussian deformation (c-f) structures dynamic receptive fields by controlling the sampling points (blue) according to the covariance. Its low-dimensionality is less general but more efficient for learning and inference than free-form deformation (b) by Dai et al. (2017), which requires more parameters.

## 5 EXPERIMENTS

We experiment with the local recognition task of semantic segmentation, because our method learns local receptive fields. As a recognition task, semantic segmentation requires a balance between local scope, to infer where, and global scope, to infer what.

**Data** CityScapes (Cordts et al., 2016) is a challenging dataset of varied urban scenes from the perspective of a car-mounted camera. We follow the standard training and evaluation protocols and train/validation splits, with $2,975$ finely-annotated training images and $500$ validation images. We score results by the common intersection-over-union metric on the validation set. We evaluate the network itself without post-processing, test-time augmentation, or other accessories to isolate the effect of receptive field learning.

**Architecture and Optimization** For backbones we choose strong fully convolutional networks derived from residual networks (He et al., 2016). Dilated residual nets (DRN) (Yu et al., 2017) have high resolution and receptive field size through dilation. Deep layer aggregation (DLA) (Yu et al., 2018) fuses layers by hierarchical and iterative skip connections. We also define a ResNet-34 backbone as a simple alternative for ablations and exploratory experiments. These are representative of common architectural patterns in state-of-the-art fully convolutional networks.

We train our models by stochastic gradient descent for $240$ epochs with momentum $0.9$, batch size $16$, and weight decay $10^{-4}$. Training follows the "poly" learning rate schedule (Chen et al., 2018; Zhao et al., 2017) with initial rate $0.01$. The input images are cropped to $800 \times 800$ and augmented by random scaling, random rotation, and random color distortions as in (Howard, 2013). We train with synchronized, in-place batch normalization (Rota Bulò et al., 2018). For fair comparison, we reproduce the DRN and DLA baselines in our same setting, which improves on their reported results.

**Baselines** The chosen DRN and DLA architectures are strong methods on their own, but they can be further equipped for learning global spatial transformations and local deformations. Spatial transformer networks (Jaderberg et al., 2015) and deformable convolution (Dai et al., 2017) learn dynamic global/local transformations respectively. Spatial transformers serve as a baseline for structure, because they are restricted to a parametric class of transformations. Deformable convolution serves as a baseline for local, dynamic inference without structure. For comparison in the static setting, we simplify both methods to instead learn static transformations.

| method | IU |
|---|---|
| DRN-A | 72.4 |
| + Extra Conv. | 72.9 |
| + STN (static) | 70.5 |
| + Deformable (static) | 72.2 |
| + Composition (ours) | **73.5** |
| + CCL | 73.1 |
| + DoG (ours) | **74.1** |
| DLA-34 | 76.1 |
| + Composition (ours) | **78.2** |

Table 1: Learning our composition improves the accuracy of careful designs.

| method | IU |
|---|---|
| ResNet-34 | 64.8 |
| + Blur | 66.3 |
| + Blur-Resample | **68.1** |
| + DoG Blur | 70.3 |
| + DoG Blur-Resample | **71.4** |
| DRN-A | 72.4 |
| + Blur | 72.2 |
| + Blur-Resample | **73.5** |

Table 2: It is better to compose with resampling than without.

| method | IU |
|---|---|
| DRN-A | 72.4 |
| w/ CCL | 73.1 |
| + Blur | **74.0** |
| w/ ASPP | 74.1 |
| + Blur | **74.3** |

Table 3: Blurring dilation helps slightly.

## 5.1 LEARNING SEMI-STRUCTURED FILTERS

**Augmenting Existing Architectures** See Table 1 for the accuracies of the architectures, baselines, and our filtering. We augment the last output stage of DRN-A with a single instance of our composition. We augment the decoder of DLA with ten instances of our composition, one at each merge layer. Optimization is end-to-end. Our composition by convolution improves by 1-2 points.

Note that these architectures are already agressively-tuned, which required significant model search and engineering effort. Our composition is still able to deliver improvement through learning without further engineering. In the next subsection, we show that joint optimization of our composition assists model search when the chosen architecture is suboptimal.

**How to Compose** We can compose with a Gaussian structured filter by blurring alone or blurring and resampling. Blurring and resampling first blurs with the Gaussian, and then transforms the sampling points for the following filtering according to the covariance. Blurring and resampling by Gaussian covariance can be interpreted as a smooth, circular extension of dilated convolution (Chen et al., 2015; Yu & Koltun, 2016) or as a smooth, affine restriction of deformable convolution (Dai et al., 2017). As either can be learned end-to-end, we experiment with both in Table 2. From this comparison we choose blurring and resampling for the remainder of our experiments.

**Blurred Dilation** To isolate the effect of blurring without learning, we smooth dilation with a blur proportional to the dilation rate. CCL (Ding et al., 2018) and ASPP (Chen et al., 2018) are carefully designed dilation architectures for context modeling, but neither blurs before dilating. Improvements from blurred dilation are reported in Table 3. Although the gains are small, this establishes that smoothing can help. This effect should only increase with dilation rate.

## 5.2 DIFFERENTIABLE RECEPTIVE FIELD SEARCH

Our composition turns choosing receptive fields into a task for optimization, instead of design. Table 4 shows how optimization counteracts the reduction of the architectural receptive field size and the enlargement of the input. These controlled experiments, while simple, reflect a realistic lack of knowledge in practice: for a new architecture or dataset, the right design is unknown.

For these experiments we include our composition in the last stage of the network. We compare fine-tuning only the last stage and end-to-end optimization. End-to-end optimization of our difference of Gaussians significantly reduces the drop in accuracy across scale shifts.

In the extreme, we can do *structural* fine-tuning by including our composition in a pre-trained network and only optimizing the covariance. When fine-tuning the structure alone, optimization either reduces the Gaussian to a delta, doing no harm, or slightly enlarges the receptive field, giving a one point boost. Therefore the special case of the identity, as explained in Figure 4, is learnable in practice.

## 5.3 DYNAMIC GAUSSIAN STRUCTURE

Dynamic inference of the covariance adaptively adjusts receptive fields to vary with the input. In these experiments we choose spherical covariance with a single degree of freedom for scale.

| method | params | epoch | IU | Δ |
|---|---|---|---|---|
| DRN-A | many | 240 | 72.4 | 0 |
| No Dilation: Smaller Receptive Field | | | | |
| ResNet-34 | many | 240 | 64.8 | -7.6 |
| + Composition | some | +20 | 68.1 | -4.6 |
| + DoG | some | +20 | 68.9 | -3.5 |
| …End-to-End | many | 240 | **71.4** | -0.8 |
| and 2× Enlarged Input | | | | |
| ResNet-34 | many | 240 | 56.2 | -16.2 |
| + Composition | some | +20 | 57.8 | -14.6 |
| + DoG | some | +20 | 62.7 | -9.7 |
| …End-to-End | many | 240 | **66.5** | -5.9 |

Table 4: Optimizing receptive fields with our composition helps adjust to the architecture and data.

| Cityscapes Validation | | | |
|---|---|---|---|
| method | dyn.? | dyn. params | IU |
| DRN-A | | - | 72.4 |
| + Gauss. Def. (ours) | ✓ | 1 | **76.6** |
| + Free-form Def. | ✓ | $2k^2$ | 76.6 |
| ResNet-34 | | - | 64.8 |
| + Gauss. Def. (ours) | ✓ | 1 | 74.2 |
| + Free-form Def. | ✓ | $2k^2$ | **75.1** |
| Cityscapes Test | | | |
| DRN-A | | - | 71.2 |
| + Gauss. Def. (ours) | ✓ | 1 | **74.3** |
| + Free-form Def. | ✓ | $2k^2$ | 73.6 |

Table 5: Gaussian deformation improves computational efficiency and rivals free-form accuracy.

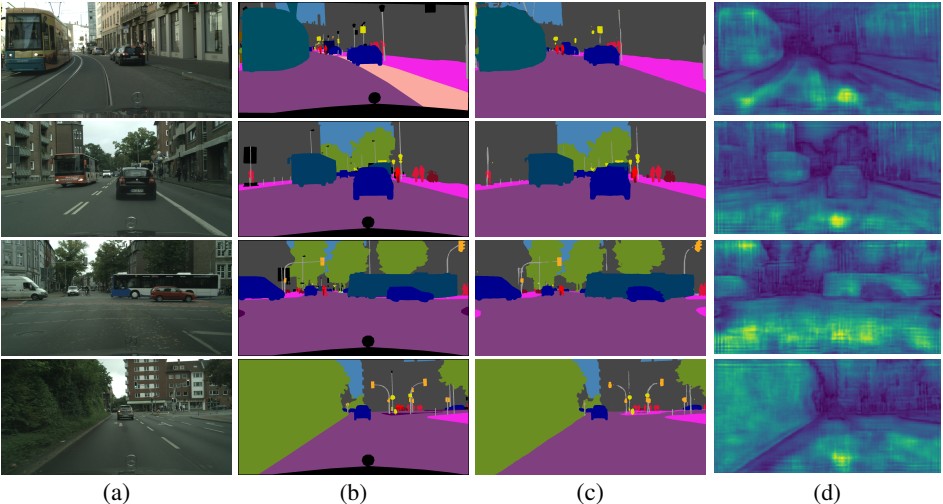

|     |     |     |     |
|:---:|:---:|:---:|:---:|
| (a) | (b) | (c) | (d) |

Figure 7: Qualitative results for dynamic scale inference: (a) inputs, (b) truths, (c) outputs, and (d) scale estimates where small is blue/dark and large is yellow/bright. The scale estimates exhibit structure: coherent segments and boundaries between them can be seen.

Qualitative results for dynamic Gaussian structure are shown in Figure 7. The local covariances reflect scale structure in the input and the output segmentation. Quantitative results compare with free-form deformation in Table 5. Gaussian structure improves efficiency while preserving accuracy.

Our results show spherical Gaussian deformation can suffice to achieve equal accuracy as general, free-form deformation. Including further degrees of freedom by diagonal and full covariance does not give further improvement on this task and data. As scale is a ubiquitous transformation in the distribution of natural images, dynamic scale inference might suffice to cope with many variations.

# 6 CONCLUSION

Composing structured Gaussian and free-form filters makes receptive fields differentiable for direct optimization of the degree of locality. Through receptive field learning, our semi-structured parameters do by gradient optimization what current free-form architectures have done by discrete design. That is, in our parameterization changes in structured *weights* would require changes in free-form *architecture*.

Our method learns local receptive fields. While we have focused on locality in space, the principle is more general, and extends to locality in time and other dimensions.

Factorization of this sort points to a reconciliation of structure and learning, through which known structure is respected and unknown detail is learned freely.

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
