# OpenReview forum: "Blurring Structure and Learning to Optimize and Adapt Receptive Fields"
_ICLR.cc/2020/Conference — Reject_

### Official Review · AnonReviewer3 · 2019-10-20
**Official Blind Review #3**

**Rating:** 3

**Review:**

Summary:
- key problem: improved visual representation learning with limited increase in parameters by leveraging Gaussian structure;
- contributions: 1) compose Gaussian blurs and free-form convolutional filters in an end-to-end differentiable fashion, 2) showing that learning the covariance enables a factorized parameter-efficient representation covering wide and flexibly structured filters, 3) experiments on CityScapes showing the proposed layers can help improve semantic segmentation performance for different architectures (DRN, DLA, and ResNet34).

Recommendation: weak reject

Key reason 1: mismatch between the generality of the claims and experiments.
- Learning to adapt and optimize receptive fields successfully would be a great fundamental improvement to CNN architectures. Experiments are done on a single dataset for a single task, which seems insufficient to support the generality of the approach and claims in the submission. I would recommend using other datasets (e.g., COCO) and tasks (e.g., object detection, instance segmentation, depth estimation/completion), where the benefits of the approach could be demonstrated more broadly and clearly (including its inherent trade-offs).
- The improved efficiency (one of the main claims) is only assessed on the number of parameters, which is a direct consequence of the parametrization. Is it significant at the scale of the evaluated architectures? Does it result in runtime performance benefits? If it is indeed a useful structural inductive bias, does it result in improved few-shot generalization performance or less overfitting? Does it enable learning deeper networks on the same amount of data?
- Why modifying only later layers in the architecture (end of 4.1)? It seems that early layers would make sense too, as it is where most of the downsampling happens.

Key reason 2: lack of clarity and details.
- Section 1 and the beginning of section 4 are repetitive and verbose; in particular, Sections 4.1 and 4.2 would benefit from less textual descriptions replaced by more concise mathematical formula (simpler in this case), especially in order to know the details behind the methods compared in Tables 1-2-3.
- Overall, the paper could contain less text describing the hypothetical advantages of the method and the basic preliminaries (section 3), to focus more on the method itself, its details and evaluated benefits. In particular, the dynamic part (section 4.2) is unclear and the method is mostly described in one sentence: "To infer the local covariances we learn a convolutional regressor, which is simply a convolutional filter." Another example of the lack of details is "many" vs. "some" in the "params" column of Table 4.
- There is also a missed opportunity to provide compelling feature visualizations and qualitative experiments (beyond Fig. 7). For instance, what are the typical covariances learned? What are the failure modes that the proposed modifications address, in particular w.r.t. thin structures and boundaries that are typical hard cases for semantic segmentation and where blurring might be counterproductive?

Additional Feedback:
- missing reference: Learning Receptive Field Size by Learning Filter Size, Lee et al, WACV'19;
- missing reference (w.r.t. local filtering): Pixel-Adaptive Convolutional Neural Networks, Su et al, CVPR'19




**Experience Assessment:**

I have published one or two papers in this area.

**Review Assessment: Checking Correctness Of Derivations And Theory:**

I carefully checked the derivations and theory.

**Review Assessment: Checking Correctness Of Experiments:**

I carefully checked the experiments.

**Review Assessment: Thoroughness In Paper Reading:**

I read the paper thoroughly.

---

> ### Author Response · Authors · 2019-11-15
> **Efficiency, the Use of Later Layers, Qualitative Results, and Further References**
>
> Thank you for the feedback, and especially for coupling each point with advice for improvement.
>
> > improved efficiency (one of the main claims) is only assessed on the number of parameters
>
> Our main claim is to make filter size differentiable and unbounded (Figures 1 & 2), and we make use of Gaussian structure to do so with parameter efficiency. The decoupling of filter size from the number of filter parameters is the point. That said, computational efficiency is important too, and relative to the use of larger kernels are method saves a significant amount of computation and memory (Figure 2 and Sec. 4.1). Relative to standard deformable convolution (Figure 6), there is a 18x reduction in memory usage going from 2*k^2 offsets to 1 spherical covariance parameter, but this is only a minor effect in the large-scale architectures in current use. With respect to sample efficiency, we do not expect the inclusion of additional Gaussian parameters to train on any less data, since by composition there is no reduction in the free-form parameters.
>
> The suggestion to explore whether Gaussian receptive fields make it possible to train more effective deeper (or shallower) nets is an interesting further direction, but we focus on characterizing their effect in already established architectures like DLA and DRN.
>
> > Why modifying only later layers in the architecture (end of 4.1)?
>
> For dynamic inference, the scale regressor needs to have sufficient receptive field itself to infer how to adjust the receptive field for the task. Sufficient receptive field is achieved by including these layers later in the network. A fuller analysis of early/intermediate/late usage would be informative for future work. Here we have concentrated on the static vs. dynamic instead.
>
> > [...] what are typical covariances learned?
> > typical hard cases [like boundaries] where blurring might be counterproductive?
>
> Figure 7 is representative of the learned dynamic covariances, in particular showing their range and how they vary within and across segments. Note that boundaries are respected in the covariance maps, in that scale can change sharply from one side to the other, and boundaries are estimated to be small. By transforming the filters, and not the input, nearby pixels can have far apart scales in this way.
>
> For learned static covariances, for instance in the DLA architecture, different covariances are learned across the skips. The deepest layer is merged with such a large covariance that it is effectively global pooling, which is of interest because the original architecture lacked a global feature (this does not hurt localization, because features from shallower layers maintain resolution).
>
> > missing reference
>
> Thank you for the further references on relevant but distinct filtering methods, which we can certainly include in the related work.
>
> - Lee et al. compose large kernels with a differentiable mask such that learning this mask controls the filter size. In contrast with our work, the mask approach requires more parameters for larger filters (as discussed in our FIgure 2), and still has a bounded maximum size equal to the kernel size.
> - Su et al. adaptively multiply filters by a fixed Gaussian kernel for spatially-varying weighting. Their filtering does not learn or adapt the size of the Gaussian, as is the focus of our work for learning receptive field size.

---

### Official Review · AnonReviewer2 · 2019-10-25
**Official Blind Review #2**

**Rating:** 6

**Review:**

This paper proposes semi-structured neural filter composed of structured Gaussian filters and the usual structure-agnostic free-form filters found in neural networks. They are optimized using end-to-end training. Effectively, this lead to increased receptive field size and shape with just few additional parameters. Further, this module is architecture agnostic and can also be integrated with any dynamic inference models. Specifically, when applied on deformable convolutional filters, the deformation at each input can be structured using gaussian filters. Empirical experiments suggest that when integrated with state-of-the-art semantic segmentation architectures, the absolute accuracy on Cityscapes improves by 2%. Large improvement in seen on naive / sub-optimal architectures for segmentation.

Given that this is first work which demonstrates the efficient composition of classic structured filters with neural layer filters, I believe that research community will benefit to good extent if this paper is accepted.

Clarification:
1. I note that single gaussian is shared across different free-form filters. Is same gaussian also shared across input channels ?
2. For dynamic inference, what is the sampling resolution used ? How is it related to diagonal elements of covariance ? 2\sigma ?
3. In case of blurring and resampling, does the model learn another filter for sampling ? To me, sampling seems similar to dynamic inference operation but with static parameters.
4. As noted in paper, blurring is fundamental hwen dilating. Does DRN-A and DLA-34 models used for comparison in Table 1 includes blurring prior to dilation ?

Additional experiment:
1. Does improved receptive field size and shape also lead to improvement in other downstream tasks such as classification, object detection, depth estimation etc. ?
2. Table 4 shows that the networks with reduced depth when integrated with composed filters can perform as well as large networks. Does this holds true when extended to above tasks ?
3. I note that in all the presented results, the composed filters are only included at the last few layers. How the results prunes out when included at the lower as well as at the intermediate layers ? Please include a plot of accuracy vs depth (at which it is included).
4. I am glad to note that Gaussian deformable models performs as good as free-form deformable models with largely reduced parameters. Can you please add total network parameters comparison in Table 5 ? Further, are these also included only at the top few layers ?
5. In Table 1, DLA-34 + DoG ?

**Experience Assessment:**

I have read many papers in this area.

**Review Assessment: Checking Correctness Of Derivations And Theory:**

N/A

**Review Assessment: Checking Correctness Of Experiments:**

I carefully checked the experiments.

**Review Assessment: Thoroughness In Paper Reading:**

I read the paper thoroughly.

---

> ### Author Response · Authors · 2019-11-15
> **Clarifying Gaussian Sharing, Sampling, and Blurring**
>
> Thank you for your feedback, and the precise clarification questions, which we address point-by-point:
>
> >  single gaussian is shared across different free-form filters. Is same gaussian also shared across input channels ?
>
> The Gaussian is shared across all input and output channels of a layer. In effect, this lets a layer learn/adapt a shared scale for all of its filters. Not sharing the Gaussians, for channel-wise scaling, is an extension for future work.
>
> > For dynamic inference, what is the sampling resolution used ?
>
> We experimented with setting the sampling rate to 2*sigma, as we did for static filtering, but found a constant sampling rate (as shown in Figure 6) to suffice in our experiments. That said, we expect that more extreme ranges of scale would require setting the resolution as a function of sigma, or else the sampling could be too sparse.
>
> > In case of blurring and resampling, does the model learn another filter for sampling ? To me, sampling seems similar to dynamic inference operation but with static parameters.
> This is exactly right. The sampling coordinates and the blurring filter are determined by the same covariance. This is analogous to smoothing and decimation when forming a pyramid: only smoothing would merely blur, but gaussian filtering then resampling/dilating the following filter instead changes scale.
>
> > blurring is fundamental when dilating. Does DRN-A and DLA-34 models used for comparison in Table 1 includes blurring prior to dilation ?
>
> Yes, but results with these architectures were not sensitive to this, since the dilation rates (2, 4) are not so large. The effect of blur was stronger for ASPP and CCL (Table 3) with larger rates (6, 12, 18).

---

### Official Review · AnonReviewer1 · 2019-10-27
**Official Blind Review #1**

**Rating:** 3

**Review:**

In this paper, the authors proposed a semi-structured composition of free-form filters and structured Gaussian filters to learn the deep representations. Experiments demonstrate its effectiveness in semantic segmentation. The idea is interesting and somewhat reasonable but I still have several concerns. However, I still have several concerns:
1.	The authors proposed to compose the free-form filters and structured filters with a two-step convolution. The authors are expected to clarify why and how the decomposition can realized its purpose? The authors need to further justify the methods by providing more theoretical analysis, and comparing with alternative methods.
2.	The experiments are rather insufficient, and the authors are expected to make more comprehensive evaluations, e.g., more comparisons with the traditional CNN models.
3.	The improvement is rather incremental compared with the alternative methods. The authors actually introduce some prior to the learning process. It would be better if the authors could show some other advantages, e.g., whether it can train the model with smaller number of samples, and whether we can integrate other prior besides Gaussian filters for other structures since Gaussian is a good prior for blurring.


**Experience Assessment:**

I have published in this field for several years.

**Review Assessment: Checking Correctness Of Derivations And Theory:**

I assessed the sensibility of the derivations and theory.

**Review Assessment: Checking Correctness Of Experiments:**

I assessed the sensibility of the experiments.

**Review Assessment: Thoroughness In Paper Reading:**

I read the paper at least twice and used my best judgement in assessing the paper.

---

> ### Author Response · Authors · 2019-11-15
> **Two-step Convolution and the Gaussian as a Prior for Learning**
>
> Thank you for pointing out the decomposition of convolution and the role of the Gaussian parameters for clarification.
>
> > authors proposed to compose the free-form filters and structured filters with a two-step convolution. [please] clarify why and how
>
> The two-step decomposition (Sec. 4.1) follows from the associativity of convolution: rather than convolve the gaussian and free-form filters then convolve the input, we can convolve the input with the gaussian and then the free-form filters. The purpose is to make use of specialized filtering for Gaussian step, in particular to use separability to reduce the complexity of filtering with a K-size filter to O(2KMN) down from O(K^2MN) for an MxN input.
>
> > authors actually introduce some prior to the learning process
>
> We do not introduce a Gaussian prior in the sense of regularizing convolutional filters to be more Gaussian. We include Gaussian filters in composition with standard free-form filters to give our networks more parameters, not fewer, for optimizing and adapting scale (Figure 1). In this sense the Gaussian is not a prior, but a different kind of parameter. We do not aim to learn from fewer samples, but instead to learn more general networks that can better handle scale differences: results show robustness to changes in architecture and data (Table 4), improved accuracy by locally adapting scale (Table 5), and qualitatively sensible scale estimates (Figure 7).

---

### Decision · Program_Chairs · 2019-12-19

**Decision:**

Reject

**Comment:**

The paper proposes an interesting idea of inserting Gaussian convolutions into ConvNet in order to increase and to adapt effective receptive fields of network units. The reviewers generally agree that the idea is interesting and that the results on CityScapes are promising. However, it is hard not to agree with Reviewer 3, that validation on a single dataset for a single task is not sufficient. This criticism is unaddressed.